# ON THE EFFECTIVENESS OF ONE-SHOT FEDERATED ENSEMBLES IN HETEROGENEOUS CROSS-SILO SETTINGS

## ABSTRACT

Federated learning (FL) is a popular approach for training machine learning models on decentralized data. For communication efficiency, one-shot FL trades the iterative exchange of models between clients and the FL server for one single round of communication. However, one-shot FL does not perform as well as iterative FL, and struggles under high data heterogeneity. While ensembles have repeatedly appeared as strong contenders in one-shot FL literature, their full potential is still under-explored. In this work, we extensively examine federated ensembles across the heterogeneity spectrum, in conjunction with various aggregation functions and a specific focus on cross-silo settings. Our experiments reveal that an aggregator based on a shallow neural network can significantly boost the performance of ensembles under high data heterogeneity. Through comprehensive evaluations on the CIFAR-10, SVHN and the cross-silo healthcare FLamby benchmark, we show that federated ensembles not only achieve up to 26% higher accuracy over current one-shot methods but can also match the performance of iterative FL under high data heterogeneity, all while being up to $9.1\times$ more efficient in terms of communication due to their one-shot nature.

## 1 INTRODUCTION

FL is a widely adopted distributed machine learning (ML) approach, enabling clients to *collaboratively train* a common model over their collective data without sharing raw data with a central server (McMahan et al., 2017). In each round of collaborative training, clients perform local model updates and send the resulting model to the central server. The server then aggregates the models and disseminates the updated global model to clients for the next round. As clients never share raw data during training, FL has been used in many applications where data privacy is critical, such as recommender systems (Hartmann et al., 2019) and medicine (Li et al., 2019). However, typical FL training tasks take thousands of communication rounds and require coordination by the central server (Bonawitz et al., 2019), which results in major communication costs (Kairouz et al., 2021).

One-shot FL addresses the communication challenges in FL by restricting the exchange of models to a single round, often by forming an ensemble of locally-trained client models at the server (Guha et al., 2019). In fact, ensembles have recurrently appeared as strong competitors in the field of one-shot FL (Guha et al., 2019; Yurochkin et al., 2019; Chen & Chao, 2021; Heinbaugh et al., 2023). However, their full potential is still under-explored, with existing work only considering simple averaging to aggregate the predictions. While other one-shot FL methods have been proposed (Yurochkin et al., 2019; Li et al., 2021), their performance is typically only evaluated against one-round FL algorithms (*e.g.*, FEDAVG). Yet, one-shot methods considerably underperform when compared to *iterative* FL. Likewise, data heterogeneity is only rarely considered in one-shot FL (Heinbaugh et al., 2023).

The potential of ensembles is even greater in *cross-silo* settings. While *cross-device* is the most common use case in FL, the numerous clients with limited data yield *weak classifiers* (Kearns, 1988), unable to produce acceptable accuracy (Lin et al., 2020), either individually or as part of an ensemble. In contrast, cross-silo settings (Kairouz et al., 2021), where a handful of large clients (*e.g.*, banks and hospitals) individually hold sufficiently large amounts of data, produce performant classifiers. At the same time, cross-silo settings are highly heterogeneous, hampering the performance

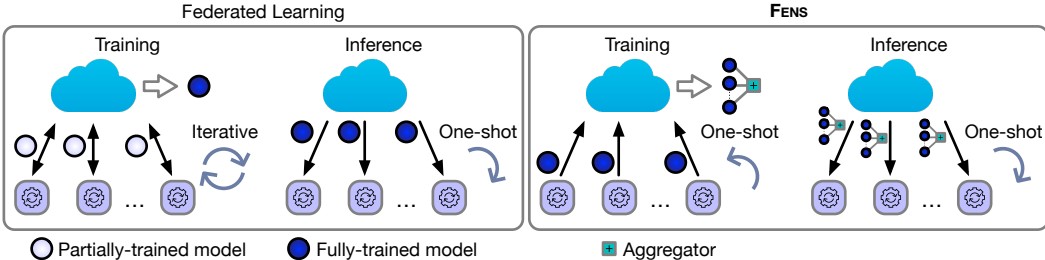

Figure 1: FENS in comparison to Federated Learning.

of FL (Ogier du Terrail et al., 2022). Interestingly, in such scenarios, putting together specialized classifiers in an ensemble might provide superior accuracy than iteratively training a single global model over heterogeneous data.

In the light of above, we thoroughly investigate Federated Ensembles (FENS) in one-shot FL (Figure 1) with various aggregation functions and varying statistical heterogeneity. Focusing on the cross-silo setting, we make the following contributions.

**Contributions.** *(i)* Our study of different aggregation functions unveils an aggregator based on a shallow neural network, which can significantly boost the performance of ensembles under high data heterogeneity. *(ii)* We perform a comparative study with several state-of-the-art iterative FL algorithms over the CIFAR-10 dataset and conclude that FENS can match their performance in the face of high data heterogeneity, while being up to $9.1\times$ more efficient in communication cost. *(iii)* We extend our analysis to two one-shot baselines — FEDKD (Gong et al., 2022) and FEDCVAE-ENS (Heinbaugh et al., 2023) — and show that FENS can achieve up to 26% higher accuracy under high data heterogeneity. *(iv)* We obtain insights into FENS by analyzing its performance according to the local dataset size. Our experiments on the SVHN datatset reaveal that, for large enough dataset sizes (which is usually the case in cross-silo settings), FENS can match the performance of iterative FL under high heterogeneity. *(v)* Lastly, we evaluate FENS on three tasks from the FLamby benchmark (Ogier du Terrail et al., 2022), a realistic cross-silo FL dataset for healthcare applications. Our results reveal that FENS achieves superior performance over one-shot FEDAVG and one-shot FEDPROX while also matching iterative FL.

## 2 RELATED WORK

**Statistical heterogeneity in FL.** Following FEDAVG (McMahan et al., 2017), several algorithms have been developed to address the challenge of data heterogeneity in FL. FEDPROX (Li et al., 2020) constrains local updates through a proximal term, while SCAFFOLD (Karimireddy et al., 2020) uses control variates to correct the client drift. FEDNOVA (Wang et al., 2020b), in turn, combines normalized gradients to tackle heterogeneous updates, whereas FEDYOGI and FEDADAM extend the adaptive optimizers from traditional ML to federated settings (Reddi et al., 2021), delivering superior performance. Other approaches tackle heterogeneity with methods that cluster statistically similar clients (Briggs et al., 2020; Liu et al., 2022) or with advanced client selection strategies (Tang et al., 2022). We remark that all these methods are designed for standard FL and therefore not feasible in one-shot FL. Nevertheless, we perform extensive empirical evaluations and show that FENS can nearly match the performance of several of the above algorithms under high statistical heterogeneity, despite its one-shot nature.

**One-shot FL.** In the context of cross-device FL, Guha et al. (2019) proposed one-shot FL, where communication is restricted to a single round. They presented two distinct approaches: *(i)* heuristic selection techniques to determine which clients are included in the final ensemble; and *(ii)* knowledge distillation (KD), which requires an auxiliary dataset for ensemble aggregation into another model at the server. After that, other methods for one-shot FL based on KD were proposed (Li et al., 2021; Gong et al., 2022). However, the additional dataset required by KD methods may not be easy to find (Zhu et al., 2021), since it must be large, publicly available, and somewhat similar to the local datasets of clients. To circumvent this issue, Heinbaugh et al. (2023) proposed a data-free one-shot FL using conditional variational autoencoders (VAEs). In their approach, called FEDCVAE-ENS, clients locally train VAEs and upload the decoder to the server, which then generates synthetic samples

from the received decoders and trains a classifier model. Apart from the above, other approaches in one-shot FL either do not fully consider statistical heterogeneity, *i.e.*, non independent and identically distributed (non-IID) data (Shin et al., 2020; Li et al., 2021), or face difficulties under high data heterogeneity (Zhou et al., 2020; Zhang et al., 2021).

Another line of research in FL that tackles the problem of aggregating fully trained client local models (Yurochkin et al., 2019; Wang et al., 2020a), also partly falls in the one-shot regime. Yurochkin et al. (2019) developed PFNM, a Bayesian non-parametric approach for fully-connected neural network architectures that matches neurons in different client models before averaging. Wang et al. (2020a) proposed FEDMA, which extends PFNM to convolutional neural networkss (CNNs) and LSTMs. However, FEDMA performs iterative layer-wise matching, requiring at least as many communication rounds as the network depth, hence not truly being one-shot.

**Ensembles in FL.** Ensembles have been previously studied in FL for a variety of different goals. Lin et al. (2020) propose FEDDF that performs robust model fusion of client ensembles to support model heterogeneity. The FEDBE algorithm (Chen & Chao, 2021) uses Bayesian Model Ensemble to aggregate client local models, improving over traditional weighted average. Hamer et al. (2020) propose the FEDBOOST algorithm, where the server has a collection of $q$ pre-trained predictors and aims to learn weights $\alpha_1, \alpha_2, \ldots, \alpha_q$ from the federated network for constructing an ensemble $\sum_{k=1}^{q} \alpha_k h_k$ that minimizes the global loss. However, these works are designed for standard FL and rely on substantial iterative communication. In the decentralized edge setting, Shlezinger et al. (2021) show that collaborative inference via neighbor averaging can achieve higher accuracy over local inference alone. However, they assume a setting where clients can exchange query data during inference and consider only IID data replicated on all edge devices.

## 3  DESCRIPTION OF FENS ALGORITHM

Consider a classification task with input space $\mathcal{X}$ and output space $\mathcal{Y}$. In this paper, we study a setting comprising $M$ clients and a trusted central server. Each client $i$ holds a local dataset $\mathcal{D}_i$ in $\mathcal{X} \times \mathcal{Y}$, which is typically different across clients, *i.e.*, data is heterogeneous. Given a parametric model $h_\theta : \mathcal{X} \to \mathcal{Z}$, each data point $(x, y) \in \mathcal{X} \times \mathcal{Y}$ incurs a loss of $\ell(h_\theta(x), y)$ where $\ell : \mathcal{Z} \times \mathcal{Y} \to \mathbb{R}$. Denoting by $\Theta \subseteq \mathbb{R}^d$ the set of possible parameters for $h_\theta$, the objective of a machine learning algorithm is to solve the empirical risk minimization (ERM) problem defined as

$$\min_{\theta \in \Theta} \quad \frac{1}{|\hat{\mathcal{D}}|} \sum_{(x,y) \in \hat{\mathcal{D}}} \ell\left(h_\theta(x), y\right) \tag{1}$$

where $\hat{\mathcal{D}} := \bigcup_{i \in [M]} \mathcal{D}_i$ is the union of the local datasets held by all clients.

### 3.1  FEDERATED LEARNING (FL)

FL algorithms, such as FedAvg (McMahan et al., 2017), can be seen as gradient-based optimization methods approximating a minimizer for the ERM. The success of these methods is mainly due to their capacity to solve the ERM problem (Equation (1)), while allowing the data to remain distributed. Essentially, an FL algorithm is an iterative method where the server updates a global parameter thanks to local computations made by the clients. Specifically, the server starts by initializing the model parameter to $\theta_0$. Then it proceeds in $T$ rounds. At each round, the server multicasts the current model parameter $\theta_t$ to a set of clients. Then, each client updates its local parameters using local training data, *i.e.*, $\theta_{t+1}^{(i)} \leftarrow \text{ClientLocalUpdate}(\theta_t; \mathcal{D}_i)$. The round finishes with each client sending their updated model $\theta_{t+1}^{(i)}$ to the server, which then updates the global parameters as

$$\theta_{t+1} \leftarrow \frac{1}{M} \sum_{i \in [M]} \theta_{t+1}^{(i)}. \tag{2}$$

### 3.2  FEDERATED ENSEMBLES (FENS)

In FENS, the iterative parameter exchanges of FL are replaced by a *one-shot* communication of locally trained models in the following procedure:

- **Local training:** Each client $i$ performs a full local training procedure to solve their own ERM problem, *i.e.*, they solve Equation (1) where $\hat{\mathcal{D}}$ is replaced by $\mathcal{D}_i$.

- **One-shot communication:** After computing an approximate solution $\theta^{(i)}$ to their local ERM problem, each client $i$ sends $\theta^{(i)}$ to the server.

- **Global model:** Upon receiving the local parameter $\theta^{(i)}$, corresponding to parametric model $\pi_i := h_{\theta^{(i)}}$, the server builds a global model $\pi \colon \mathcal{X} \to \mathcal{Z}$ of the form

$$\pi = f(\pi_1, \dots, \pi_M) \tag{3}$$

where $f : \mathcal{Z}^M \to \mathcal{Z}$ is an aggregation function.

Accordingly, the design of FENS consists in choosing the aggregation function $f$ in the global model computation (Equation (3)). If proxy data $\mathcal{D}_{\text{proxy}}$ is available at the server, this function $f$ can be *learned* on $\mathcal{D}_{\text{proxy}}$ using the local models $\pi_1, \dots, \pi_M$. Otherwise, if such data is unavailable, $f$ can be chosen *independently* of the local models, *e.g.*, by averaging. In Section 3.3, we give concrete examples of aggregation methods.

**Communication cost analysis.** To compare the communication cost of FENS against that of iterative FL, we take into account both training and inference[1] costs. For FENS, the local models $(\pi_1, \dots, \pi_M)$ along with the aggregation $f$ constitute the global model (Figure 1). The cost of uploading the trained models to the server in one shot entails the training communication costs for FENS. For local inference, the models and aggregation are sent back to clients, which constitute the communication costs associated with inference.

Table 1: Comparison of communication costs between FENS and iterative FL for a total of $M$ clients. Given that in cross-silo FL $M \ll T$, FENS incurs significantly lower total communication w.r.t. FL.

| PHASE | FL | FENS |
|---|---|---|
| Training | $2dMT$ | $dM$ |
| Inference | $dM$ | $|f|M + dM^2$ |
| Total | $\mathcal{O}(dMT)$ | $\mathcal{O}(dM^2)$ |

In contrast, the training phase of FL consists of several rounds of model exchange with the server, each of significant size, depending on the model dimension $d$. Indeed, over $T$ rounds of communication, the server and clients exchange $2 \times T \times M$ messages, each of size $d$. However, FL inference cost is cheaper, since only the single final model needs to be shipped to each client. Essentially, FENS trades the huge training communication cost of FL for a slightly larger inference cost. We detail these costs in Table 1. Symbols $d$ and $|f|$ indicate the size of the local models and aggregation model, respectively. $T$ is the total number of communication rounds for FL.

## 3.3 FENS AGGREGATION METHODS

For performing the ensemble aggregation, we have different choices for $f$ in Equation (3). We classify these methods into *trainable* and *non-trainable* methods and evaluate them in Section 4. For simplicity, we consider $\mathcal{Y} = [C]$, *i.e.*, multi-class classification with $C$ classes, and $\mathcal{Z}$ to be the probability simplex over $\mathcal{Y}$. Thus, each model $\pi_i$ maps a data point to a probability distribution over $[C]$, *i.e.*, a $C$-dimensional vector.

**Trainable methods.** Trainable methods correspond to parametric aggregations trained on proxy data available at the server, using the local models $\pi_1, \dots, \pi_M$. The proxy data is commonly assumed to be available at the server and is used by FL model engineers to bootstrap tasks or perform initial hyperparameter exploration (Bonawitz et al., 2019). This dataset could be derived from publicly available data, a portion of clients' data that is not privacy-sensitive, or be synthetically generated (Kairouz et al., 2021). Trainable methods can be generically formulated as the search for parametric model $f_{\boldsymbol{\lambda}}$ which solves the ERM Problem (Equation (4)) for $\boldsymbol{\lambda}$ in parameter space $\Lambda \subseteq \mathbb{R}^q$ on proxy dataset $\mathcal{D}_{\text{proxy}}$:

$$\min_{\boldsymbol{\lambda} \in \Lambda} \quad \frac{1}{|\mathcal{D}_{\text{proxy}}|} \sum_{(x,y) \in \mathcal{D}_{\text{proxy}}} \ell\left(f_{\boldsymbol{\lambda}}(\pi_1(x), \dots, \pi_M(x)), y\right). \tag{4}$$

---

[1]In this context, we use the term "inference" to refer to the costs incurred when deploying FENS or FL after the training phase is complete. This should not be confused with the cost associated with performing a single inference, *i.e.*, labeling unknown data.

As such, the learned aggregation function $f_\lambda$ can take the form of a neural network, or a linear model as in (Hamer et al., 2020). Recall that the communication cost of FENS in the deployment phase is driven by $|f_\lambda| = q$. In fact, we show that a shallow neural network suffices for $f_\lambda$ for several tasks. As a consequence, $q$ can be significantly smaller than $d$. The additional communication cost of transferring the aggregator is therefore minimal. We also explore trainable methods inspired from voting theory such as *polychotomous voting* (Ben-Yashar & Paroush, 2001) (Appendix C.2).

**Non-trainable methods.** Non-trainable methods correspond to static aggregation rules like averaging, median, weighted averaging *etc.*, commonly used in ensemble learning (Polikar, 2006; Ganaie et al., 2022). Of particular interest to us is the *weighted averaging* method (Gong et al., 2022), which we generically formulate as

$$f(\pi_1, \ldots, \pi_M; \lambda_1, \ldots, \lambda_M) = \sum_{i=1}^{M} \lambda_i \odot \pi_i \qquad (5)$$

where $\lambda_1, \ldots, \lambda_M \in [0, 1]^C$ are weight vectors and $\odot$ denotes coordinate-wise product. Plain averaging corresponds to $\lambda_i = [\frac{1}{M}, \ldots, \frac{1}{M}]$. In particular, this method can use the number of samples of a given class that a client possesses in order to weigh the probabilities. However, this requires each client to share its label distribution, which may be privacy-invasive in some applications.

## 4 EXPERIMENTS

We split our evaluation into the following sections: *(i)* the CIFAR-10 dataset (Krizhevsky et al., 2014) to evaluate FENS in comparison to FL algorithms (Section 4.1); *(ii)* the SVHN (Netzer et al., 2011) dataset to assess the impact of data size w.r.t. data heterogeneity in FENS and FEDAVG (Section 4.2); and *(iii)* the realistic cross-silo FLamby (Ogier du Terrail et al., 2022) benchmark (Section 4.3).

### 4.1 CIFAR-10

#### 4.1.1 DATA PARTITIONING

For all experiments in this section, we consider a setting with 20 clients on the CIFAR-10 dataset. Its original training and testing sets consist of 50 000 and 10 000 samples, respectively. We split the original training set into 90-10% to respectively obtain $\mathcal{D}_{\text{train}}$ and $\mathcal{D}_{\text{proxy}}$, whereas the testing set is split (50-50%) into $\mathcal{D}_{\text{test}}$ and $\mathcal{D}_{\text{val}}$. We use $\mathcal{D}_{\text{val}}$ to tune hyperparameters and always report the accuracy on $\mathcal{D}_{\text{test}}$. To obtain the client datasets, we partition $\mathcal{D}_{\text{train}}$ using the Dirichlet distribution $\text{Dir}_{20}(\alpha)$, as done in previous work (Wang et al., 2020b; Gong et al., 2022). The parameter $\alpha$ determines the degree of heterogeneity, with lower values indicating non-IID data distributions (see Appendix A, Figure 5). The $\mathcal{D}_{\text{proxy}}$ split is located at the server and used by FENS to train the aggregator. For FL algorithms, we partition the entire training set ($\mathcal{D}_{\text{train}} \cup \mathcal{D}_{\text{proxy}}$) across clients. This enables a fair comparison between FL and FENS, both seeing equal amounts of training data. Similarly, non-trainable FENS methods (Section 3.3) also partition $\mathcal{D}_{\text{train}} \cup \mathcal{D}_{\text{proxy}}$ across clients.

#### 4.1.2 SETUP AND HYPERPARAMETERS

We use ResNet-8 (He et al., 2016) as the model and a fixed batch size of 16 across all our experiments. Other parameters for each setup are described below.

**FENS methods.** Each client in FENS performs local training for 250 epochs using SGD as the local optimizer. The learning rate is set to $2.5 \times 10^{-3}$ and decayed using Cosine Annealing. For the neural network (NN) aggregation method, we use a simple multilayer perceptron with 1 hidden layer comprising of 120 units each, using ReLu activations and a final softmax output layer. The NN aggregator is trained using the ADAM optimizer for 300 epochs with a learning rate of $5 \times 10^{-5}$. It further adopts an exponential decay with the parameter $\gamma = 0.999$. Parameters for other aggregation methods presented in Section 3.3 are described in Appendix B.1.

**Baseline FL algorithms.** For comparison with FL, we consider 6 algorithms, including the standard FEDAVG (McMahan et al., 2017); the heterogeneity-robust FEDPROX (Li et al., 2020), FED-NOVA (Wang et al., 2020b) and SCAFFOLD (Karimireddy et al., 2020); and the adaptive optimizer-based FEDYOGI and FEDADAM (Reddi et al., 2021). We tune learning rates for each algorithm

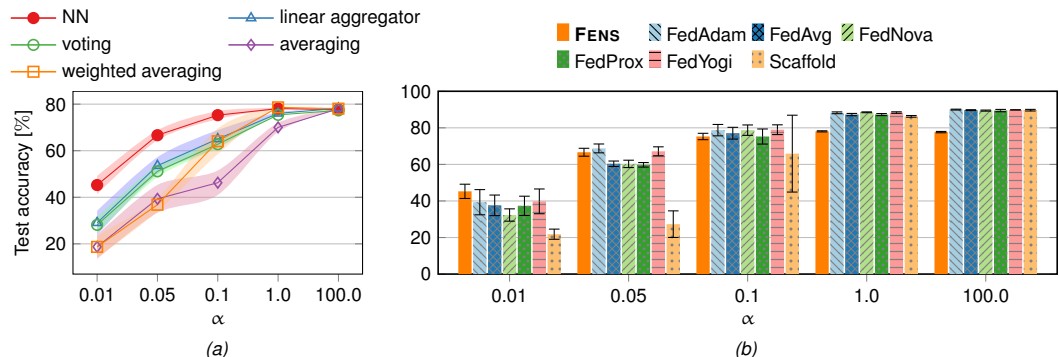

Figure 2: **(a) FENS aggregation methods:** NN performs the best. **(b) FENS vs FL:** FENS achieves competitive accuracies when the heterogeneity is very high ($\alpha \leq 0.1$). However, FL is significantly better when the data tends to be independent and identically distributed (IID) ($\alpha \geq 1.0$). Numerical values for all plots are included in Tables 8 and 9 (Appendix D) for reference.

and train for 100 communication rounds (details in Appendix B.2). The server assumes full client participation, *i.e.*, all 20 clients participate in each round, which is common in cross-silo FL with a limited number of clients. Each client performs 2 local epochs per round. While we run 100 communication rounds, we found all algorithms to converge much before the $100^{th}$ round.

**One-shot FL baselines.** We compare FENS against three one-shot baselines: the auxiliary dataset based FEDKD (Gong et al., 2022), data-free FEDCVAE-ENS (Heinbaugh et al., 2023) and the standard one-round FEDAVG. For one-shot FEDKD, we use the setup described by Gong et al. (2022) without inference quantization, using CIFAR-100 as the public dataset for distillation. For FEDCVAE-ENS, we use the same CVAE as well as the CNN classifier from Heinbaugh et al. (2023). Moreover, we experiment with ResNet-8 as the classifier. We run all experiments using the best-reported hyperparameters for the CIFAR-10 task in Heinbaugh et al. (2023). In addition to these one-shot baselines, we implement FEDAVG with gradient compression, following the sparsification and quantization schemes of STC (Mora et al., 2022). In particular, we set the quantization precision and sparsity level to match the communication cost of FENS (details in Appendix C.1) and the remaining setup to the same as FL.

### 4.1.3 RESULTS

Each experiment 5 ran times with different seeds and we report results with a 95% confidence interval.

**FENS aggregation methods.** Figure 2(a) depicts the accuracy of all aggregation methods for 5 values of $\alpha$, showing a predictable increase as heterogeneity decreases. Remarkably, all methods converge to a single accuracy of $\sim$78% at $\alpha = 100$. The two trainable methods of voting and linear aggregation perform similarly. However, NN significantly outperforms all other methods when heterogeneity is high, suggesting the importance of non-linearity for effective aggregation. Although both plain and weighted averaging underperform at high heterogeneity, the latter quickly improves as heterogeneity decreases. Given these results, we pick the NN approach for comparison against the FL methods.

**FENS vs FL.** Figure 2(b) charts the test accuracy of FENS against the FL algorithms across the heterogeneity spectrum. Under extreme heterogeneity ($\alpha = 0.01$), FENS surpasses all FL algorithms, and by around 6% in accuracy for the best performing FEDYOGI. Similarly, at very high heterogeneity ($\alpha = 0.05$), FENS remains competitive to FEDYOGI and FEDADAM while surpassing all the others. The accuracy values at $\alpha = 0.1$ are also similar. Beyond this point, FL demonstrates better performance than FENS. In nearly IID data settings ($\alpha \geq 1.0$), FL algorithms achieve 12% higher accuracy than FENS. This happens because iterative FL thrives when learning from numerous clients with similar data distributions. However, when the local data distributions differ, the global model struggles to learn shared characteristics. In such scenarios, FENS can effectively harness diverse local classifiers, thanks to the NN aggregator. We investigate this phenomenon further in Section 4.2.

We analyze communication costs in Table 2 for $\alpha = \{0.1, 1\}$ by computing the number of rounds required to reach a pre-defined target accuracy. We set the target accuracy as the one achieved by

FENS. Although FL attains superior final accuracy at $\alpha = 1$, we still compare the corresponding differences in communication cost for scenarios where a lower target might be acceptable. FENS notably reduces total communication costs compared to FL, despite increased inference costs. Overall, FENS can save $6.4 - 9.1\times$ communication when the data is heterogeneous and $2.5 - 3.5\times$ under IID data.

Table 2: FENS vs FL communication costs comparison on the CIFAR-10 dataset. Training, inference, and total costs are presented in gibibytes (GiB).

| Algorithm | $\alpha = 0.1$ (non-IID), Target = 75.2% | | | | | $\alpha = 1$ (IID), Target = 77.6% | | | | |
|---|---|---|---|---|---|---|---|---|---|---|
| | Rounds | Train | Inference | Total | Factor | Rounds | Train | Inference | Total | Factor |
| FEDAVG | 85 | 62.21 | 0.37 | 62.58 | **8.5×** | 34 | 24.88 | 0.37 | 25.25 | **3.4×** |
| FEDPROX | 91 | 66.60 | 0.37 | 66.97 | **9.1×** | 35 | 25.61 | 0.37 | 25.98 | **3.5×** |
| FEDYOGI | 64 | 46.84 | 0.37 | 47.21 | **6.4×** | 27 | 19.76 | 0.37 | 20.13 | **2.7×** |
| FEDADAM | 75 | 54.89 | 0.37 | 55.26 | **7.5×** | 25 | 18.29 | 0.37 | 18.66 | **2.5×** |
| SCAFFOLD | 92 | 67.33 | 0.37 | 67.70 | **9.2×** | 35 | 25.61 | 0.37 | 25.98 | **3.5×** |
| FEDNOVA | 64 | 46.84 | 0.37 | 47.21 | **6.4×** | 25 | 18.29 | 0.37 | 18.66 | **2.5×** |
| **FENS** | **1** | **0.37** | **6.95** | **7.32** | – | **1** | **0.37** | **6.95** | **7.32** | – |

**FENS vs one-shot FL.** Table 3 provides a comprehensive evaluation of test accuracy of FENS against one-shot FL baselines and gradient compression in FL. We observe that all one-shot methods struggle under high heterogeneity where FENS consistently outperforms the baselines. FEDCVAE-ENS exhibits consistent accuracy across different heterogeneity levels. However, it remains sensitive to the choice of classifier architecture and struggles to perform well on CIFAR-10, despite its strong performance on simpler tasks like MNIST (Heinbaugh et al., 2023). In homogeneous data distributions, both one-shot FEDKD and FENS exhibit similar performance. However, in more heterogeneous settings, FENS outperforms one-shot FEDKD by a substantial margin, up to 26% higher accuracy at $\alpha = 0.05$. Furthermore, FENS consistently delivers superior results compared to gradient compression when operating under similar communication budgets.

Table 3: FENS vs one-shot FL algorithms on CIFAR-10.

| Algorithm | Test Accuracy [%] | | | | |
|---|---|---|---|---|---|
| | $\alpha = 0.01$ | $\alpha = 0.05$ | $\alpha = 0.1$ | $\alpha = 1$ | $\alpha = 100$ |
| FEDAVG one-round | 10.52±1.03 | 16.78±4.88 | 10.34±1.16 | 15.74±1.99 | 17.49±1.72 |
| FEDCVAE-ENS (ResNet-8) | 21.54±3.92 | 25.83±1.81 | 29.07±1.61 | 27.86±1.38 | 26.62±1.16 |
| FEDCVAE-ENS (default CNN) | 32.85±1.23 | 32.10±2.05 | 32.90±0.97 | 31.22±1.49 | 29.51±0.73 |
| FEDKD | 23.07±4.40 | 40.12±5.18 | 60.48±13.90 | **78.43±1.21** | **79.41±0.27** |
| Grad. Compression | 34.5±0.92 | 40.54±1.33 | 57.64±0.66 | 71.06±0.44 | 73.68±0.13 |
| FENS-NN (ours) | **45.26±4.45** | **66.65±2.47** | **75.27±1.99** | 78.08±0.35 | 77.67±0.42 |

## 4.2 Understanding the performance of FENS

In this section, we aim to assess the performance of FENS under high data heterogeneity. As noted earlier, FL struggles to generate a good global model when the local datasets of clients significantly differ. In such cases, an ensemble of models, each specialized on data belonging to distinct clients, shows better performance as demonstrated by our results on CIFAR-10 (Section 4.1). The quality of ensembles is highly dependent on the quality of local models, which in turn depends on the amount of local data held by the clients. As local models improve at generalizing locally, the overall performance of the ensemble is enhanced. On the other hand, more volume of data does not analogously benefit FL due to high data heterogeneity. We validate this intuition through the following experiments on the SVHN dataset.

### 4.2.1 Setup & Hyperparameters

We study the performance of FL and FENS by progressively increasing the volume of data held by the clients. To this end, we consider the classification task on the SVHN dataset due to the availability

of an extended training set of $604\,388$ samples, *i.e.*, more than $10\times$ bigger than CIFAR-10. We then experiment with fractions ranging from 10 to 100% of the total training set. For each subset, we split it into 90-10% to obtain $\mathcal{D}_{\text{train}}$ and $\mathcal{D}_{\text{proxy}}$, respectively. We then compare FENS with FEDAVG (FL baseline) on two levels of heterogeneity: $\alpha = 0.1$ (non-IID) and $\alpha = 1.0$ (IID). We tune the learning rate for FEDAVG (details in Appendix B.3) and keep the remaining setup as in previous experiments.

### 4.2.2 RESULTS

Figure 3 shows the results and confirms our prior insight behind the effective performance of FENS. Specifically, we observe that the growing volume of training data benefits FENS much more than FEDAVG. When the data distribution is homogeneous ($\alpha = 1$), the performance of FENS improves faster than FEDAVG, but still remains behind. On the other hand, under high heterogeneity ($\alpha = 0.1$), FENS quickly catches up with the performance of FEDAVG, matching the same accuracy when using the full training set. In summary, our observations can be attributed to the deterioration in FL performance, coupled with the efficacy of ensembles in harnessing heterogeneous models in the face of significant data heterogeneity.

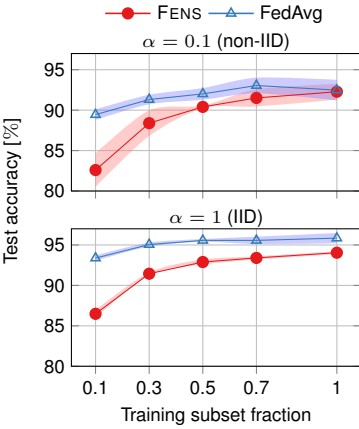

Figure 3: Performance of FENS with increasing number of samples.

### 4.3 FLAMBY MEDICAL BENCHMARK

In this section, we evaluate the performance of FENS on the real-world cross-silo FLamby benchmark (Ogier du Terrail et al., 2022).

### 4.3.1 DATASETS AND TASKS

The FLamby suite consists of seven datasets with a diverse set of tasks, including 3D segmentation, survival analysis, binary and multi-class classification. Being a cross-silo FL benchmark, the number of clients in each dataset is relatively low and varies between two and six. Owing to the large size of medical data (scans, slides *etc.*), datasets are quite large (up to $850\,\text{GiB}$) and take significantly long preprocessing and training times. We focus on the classification tasks and experiment on 3 datasets: Fed-Camelyon16, Fed-Heart-Disease, and Fed-ISIC2019. Table 4 (Appendix A) presents an overview of the selected tasks. The datasets consist of a natural non-IID partitioning across clients. We reserve 10%, 15%, and 15% of the client datasets as $\mathcal{D}_{\text{proxy}}$ at the server for Fed-Heart-Disease, Fed-Camelyon16, and Fed-ISIC2019, respectively. In other words, FENS clients train on the portion of local data not in $\mathcal{D}_{\text{proxy}}$ while the server trains the NN aggregator using $\mathcal{D}_{\text{proxy}}$. We note that clients in other FL algorithms use 100% of their local dataset for training. This ensures a fair comparison such that all methods observe the same amount of training data. Finally, all performances are reported on the original testing split $\mathcal{D}_{\text{test}}$.

### 4.3.2 SETUP AND HYPERPARAMETERS

In FLamby, the authors benchmark various FL algorithms (*strategies*), including FEDAVG, FEDPROX, FEDYOGI, and SCAFFOLD, and compare them to the *pooled training* and client local baselines. While these benchmarks were obtained after extensive tuning of hyperparameters, the authors purposefully restricted the number of rounds to be approximately the same as the number of epochs required to train on pooled data (see Ogier du Terrail et al. (2022)). Since this might not reflect true FL performance, we rerun all FL strategies to convergence using the reported tuned parameters. Precisely, we run up to $10\times$ more communication rounds than evaluated in the FLamby benchmark. For FENS, each client performs local training with the same hyperparameters as the client local baselines in FLamby while the server trains the NN aggregator using a similar configuration as before. For the one-shot FEDAVG and FEDPROX baselines, we additionally tune the number of local updates. We remark that FEDKD is infeasible in these settings since it requires a public dataset for distillation, unavailable in the medical setting. FEDCVAE-ENS is also infeasible due to the difficulty in learning good decoders for medical input data, a conclusion supported by its poor performance on the comparatively simpler CIFAR-10 task (Table 3).

### 4.3.3 RESULTS

Figure 4 shows the results with the first row comparing FENS against iterative FL algorithms and the second row against one-shot FL and the client local baselines. When comparing to iterative FL, we observe that FENS is on par for the Fed-Heart-Disease dataset and performs better for Fed-Camelyon16. However, the iterative FL algorithms achieve significantly higher accuracy on the Fed-ISIC2019 dataset. We justify this poor performance of FENS with some key observations on the client local baselines.

For both Fed-Camelyon16 and Fed-Heart-Disease datasets, the client local baselines perform well, while FL performance is affected by high data heterogeneity. The deterioration is more significant for Fed-Camelyon16, which learns on large brain slides ($10000 \times 2048$), than for Fed-Heart-Disease, which learns on tabular data. In such scenarios, FENS can harness diverse local classifiers to attain good performance. In contrast, in the case of Fed-ISIC2019, the clients possess highly unbalanced quantities of data samples (Table 4, Appendix A), producing

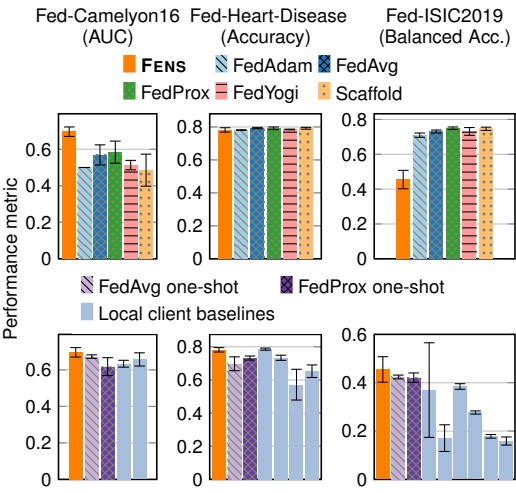

Figure 4: **FENS in FLamby.** FENS is on par with iterative FL (row-1), except when local models are weak (Fed-ISIC2019) while remaining superior in the one-shot setting (row-2). Numerical results included in Tables 10 to 15 (Appendix D).

weak local classifiers. Thus, FENS struggles to keep up with the performance of iterative FL, matching our intuition from Section 4.2. However, we note that FENS achieves superior performance over one-shot FEDAVG and one-shot FEDPROX, while performing at least as well as the best client local baseline. Overall, these observations for FL algorithms have spurred new interest in developing a better understanding of performance on heterogeneous cross-silo datasets (Ogier du Terrail et al., 2022). We show that FENS remains a strong competitor in such settings.

## 5 DISCUSSION AND CONCLUSION

**Limitations.** One limitation is the privacy aspect, as client models are shared and not aggregated. This leaves room for attacks like membership inference (Melis et al., 2019). Future work can explore mitigation techniques such as differential privacy (Geyer et al., 2017) or trusted execution environments (Messaoud et al., 2022) to address this concern. Another limitation is resource usage at inference time, as FENS requires additional memory and compute resources to store and process the complete global model. However, this is more manageable in cross-silo federated setups with reliable and powerful clients.

**Benefits.** In addition to the benefits of low communication costs and rapid training, FENS provides three important advantages. Firstly, it supports model heterogeneity, allowing for different model architectures across federated clients, which is beneficial when organizations or entities have specific requirements or constraints for their models (Li & Wang, 2019). Secondly, FENS aligns with the design principles of the popular SISA framework (Bourtoule et al., 2021) for practical machine unlearning, towards the goal of the *right to be forgotten* in GDPR (Mantelero, 2013). This enables efficient unlearning compared to traditional FL where unincorporating knowledge from a single global model can be very costly. Lastly, FENS enables model reusability by allowing clients to contribute previously trained models Li et al. (2021), fostering collaboration without the need to start training from scratch. This approach prevents resource wastage and lowers entry barriers for clients.

To conclude, this paper investigates Federated Ensembles (FENS), with a specific focus on cross-silo federated settings. FENS emphasizes local training and one-shot model sharing, thereby reducing communication costs and accelerating training. The experiments on a wide range of tasks demonstrated that FENS with the NN aggregator is notably effective in settings with high data heterogeneity, reducing the communication costs by a factor of up to $9.1\times$. Furthermore, FENS offers additional benefits of model heterogeneity, machine unlearning, and model reusability.

REPRODUCIBILITY STATEMENT

We have undertaken several steps to ensure the integrity, reproducibility and replicability of FENS. We have provided an extensive description of FENS in the Section 3, describing formally its algorithmic procedure in Section 3.2 and Section 3.3. To facilitate the reproducibility of the experimental results, the complete source code used for the evaluation of FENS will be made publicly available and a link will be added in the final paper version. We have used publicly available ML models and datasets. Each presented evaluation carries a subsection on setup and hyperparameters (Sections 4.1.2, 4.2.1 and 4.3.2) with sufficient information to reproduce all our results. We believe that these efforts will aid researchers in understanding, replicating, and building upon FENS.

ETHICS STATEMENT

We affirm that the medical datasets employed in our experiments within the FLamby benchmark were acquired and utilized in strict accordance with their respective licensing agreements and ethical guidelines. We obtained the necessary permissions and approvals from the appropriate authorities and/or institutions responsible for data collection, and we adhered to all relevant ethical standards and regulations.

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
