# OpenReview forum: "On the Effectiveness of One-Shot Federated Ensembles in Heterogeneous Cross-Silo Settings"
_ICLR.cc/2024/Conference — Submitted to ICLR 2024_

### Official Review · Reviewer_ne5T · 2023-10-25

**Soundness:** 3 good
**Presentation:** 3 good
**Contribution:** 2 fair
**Rating:** 3
**Confidence:** 5

**Summary:**

The authors investigate in which situations iterative FL is truly needed wrt using different ensembling techniques (so-called one-shot FL) while focusing on the cross-silo FL setting. The authors identify scenarii with both toy datasets (SVHN and CIFAR-10) and realistic ones (FLamby) where ensembling is sufficient to match or outperform iterative FL.

**Strengths:**

The paper highlights a question that is interesting and still opens to this day: is it always worth going through all the troubles to do FL ? Can ensembles be sufficient ? Especially in medical settings where privacy is an issue.
The paper moves progressively from developing an intuition using toy datasets and then testing/validating this intuition on a more realistic dataset.
Experiments on data starvation and changing heterogeneity are interesting
The paper writing is fair and the paper is clear.

**Weaknesses:**

Major:
- The model fusion strategy labeled as FENS is a specific case of what we call "late fusion" in the multimodal ML literature (see i.e. the references 39 and 40 in [1]), which has been already explored in depth (see all the rich literature on various fusion modules Multimodal Low-rank Bilinear pooling (MLB) / Multimodal Compact Bilinear (MCB), Multimodal Tucker Fusion, Attention, etc.).
The authors should orient the paper more towards examining late fusion schemes vs FL instead of proposing a new method because the method is far from new. In this respect the title of the article is fine: the focus should be on comparing ensembles to (iterative) FL.
The reviewer encourages the authors to drop the term FENS and to either remove or considerably shorten the corresponding sections 3.2 and 3.3.
- It is very mysterious for the reviewer the impact of having access to Dproxy in the server. The reviewer questions even adding Dproxy in the first place as the question tackled by the paper should be whether ensembling methods can match FL and with the current framing it becomes: "what is the effect of post-training on another dataset ?".  If the authors want to use this Dproxy anyway they should add a comparison with an (iteratively)-FL trained model fine-tuned on Dproxy. Note that some concerns are also linked to the two first questions of the reviewer. Authors should change Figure 2b) to compare the best FL method (aka FedAdam) vs the best trained and the best untrained ensembling methods vs the best FL method fine-tuned on Dproxy.
- Experiments on FLamby are inconclusive at best FENS does seem to outperform FL methods on Camelyon16 for Fed-Heart Disease and Fed-ISIC it is not the case so that goes a bit in the opposite direction that the paper argues. Note that Fed-IXI and Fed-TCGA-BRCA are also part of FLamby and relatively small the reviewer would like experiments on such datasets in order to form a more complete picture. Figure 4 would also benefit from including non-trained aggregation methods.
- All the discussion on communication efficience Table 1 and 2 is wasteful. By design, ensembles will require order of magnitudes less communications no need to hammer the point with two tables at least they should go in the supplementary. If the authors want to keep it at least they should compare to efficient (sparse and quantified) iterative FL approach but the reviewer thinks this is not a very interesting thing to add to the paper.
- So many more experiments would be needed to make this paper into a great paper. The reviewer already mentioned other FLamby datasets but also different experiments varying the heterogeneity source between centers by introducing artificial spurious correlations would be interesting (aka in once center all SVHN images are red and the center has a majority of one digit). Are there scaling laws for FL to be preferred to ensembles, aka starting from which critical mass does FL become interesting ? Does it depend on the task ? Is there a good formula to predict the gap between FL and ensembles from say #samples in local centers and alpha ? Does this formula transfer to realistic settings ? What about differential privacy ? For a fixed privacy budget is it better to do ensemble ? This is such an interesting question and there are so few experiments in the paper !

Minor

- The argument on model reusability in the conclusion should be dropped nothing prevents from reusing an iteratively trained model. The argument on easier unlearning for ensembles is a bit far-fetched
- Fed-Camelyon16 learns on small (and not large) ResNet features extracted from large breast slides (and not brain !). Features are larger than traditional clinical data (relative) but not large (asolute)
- The reviewers encourage the authors to insist on the privacy aspect of the ensemble methods because the two real gains of FENS in medical settings is easier IT setup and contractualization and better privacy.


Summary: the authors study a great question but do not provide enough high-quality experiments and ablation-study for it to be sufficiently interesting for ICLR.




[1] Liu, Kuan, et al. "Learn to combine modalities in multimodal deep learning." arXiv preprint arXiv:1805.11730 (2018).

**Questions:**

- It is unclear to the reviewer whether Dproxy is contained in the local dataset used in FL for CIFAR-10/SVHN and FLamby experiments. Even for FLamby the sentence: "We note that clients in other FL algorithms use 100% of their local datasets for training" is unclear.
- Are error bars averaged across different random selection of Dproxys ?

See Weaknesses for additional experiments to include to strenghten the submission.

---

> ### Author Response · Authors · 2023-11-15
> **Response to Reviewer ne5T**
>
> We kindly thank the reviewer for their thorough evaluation and constructive feedback on our manuscript. We appreciate the time and effort you have dedicated to assessing our work and we are happy to clarify some of the concerns below.
>
> **1) The model fusion strategy labeled as FENS is a specific case of what we call "late fusion" in the multimodal ML literature.**
>
> → We thank the reviewer for the references and acknowledge the similarities to late fusion in multimodal ML literature. However, in this work, we did not intend to propose new methods but extensively assess existing ensemble techniques towards the goal of understanding whether one-shot FL can match iterative FL. Throughout our work, we have made an honest attempt to acknowledge that our assessed methods are not newly proposed but derive from existing ensemble literature. We will further clarify this in the final version of the paper and look forward to orienting the description of FENS in the preview of late fusion techniques.
>
> **2) It is very mysterious for the reviewer the impact of having access to Dproxy in the server.**
>
> → We acknowledge that leveraging D_proxy at the server is atypical. Indeed, while training the aggregator can be viewed as a separate post-training step, we would like to point out that such training is considerably simpler than, for instance, training another global model through distillation at the server in FedKD. The aggregator NN operates on a very small input size (n_clients x n_classes = 20 x 10 in CIFAR-10) and has less than 4500 parameters (MLP with 1 hidden layer), allowing training even with a modest or small D_proxy dataset.
> To ensure a fair comparison, both FENS and traditional FL methods learn from the same amount of training data. In FENS, clients train on distributed D_train and learn the aggregator on D_proxy, while iterative FL clients train on distributed splits of D_train U D_proxy. Regarding your suggestion for an additional baseline, due to constraints of the rebuttal process, we might encounter limitations in the time available to thoroughly assess all aspects. Nevertheless, we will carefully consider this suggestion during the final revisions.
>
> **3) All the discussion on communication efficience Table 1 and 2 is wasteful.**
>
> → We included the communication table to highlight FENS's inference cost, which is notably higher than traditional FL algorithms due to ensemble transfer. Our comprehensive analysis in Table 3 reveals that, even with added inference costs, FENS maintains significantly lower total communication costs. We also compared FENS to sparse and quantified FL (Grad. Compression) in Table 3, by setting the quantization and sparsification level to match FENS's communication costs (Table 7). Unfortunately, we couldn't include this baseline in Table 2, as it didn't reach our target accuracy for computing communication costs.
>
> **4) Experiments on FLamby are inconclusive.**
>
> → While FENS doesn't outperform FL on the Fed-ISIC task, our aim wasn't to assert FENS as strictly superior to iterative FL which would be an inaccurate conclusion. Instead, we sought to understand the conditions favoring or hindering FENS. As discussed in Section 4.2, our intuition justifies the subpar Fed-ISIC performance by linking it to the quality of client models in the ensemble. Figure 4, row 2, illustrates notably weaker client local baselines for Fed-ISIC, contributing to FENS's suboptimal performance. In such instances, traditional FL excels in leveraging client local data through iterative communication.
>
> **5) So many more experiments would be needed to make this paper into a great paper.**
>
> → We appreciate the reviewer's insightful comments on potential research questions. While privacy concerns were not explicitly addressed in our current work, we recognize their importance and believe they warrant a separate and comprehensive investigation. In our experiments, we covered diverse scenarios, demonstrating FENS outperforming (Fed-Camelyon), matching (Fed-Heart-Disease), and underperforming (Fed-ISIC) iterative FL. Our work contributes novel empirical insights, revealing that one-shot methods can rival iterative FL under high data heterogeneity (Figure 2b). Additionally, we highlighted distinct scaling laws for FL and FENS, showcasing FENS's performance improvement with larger local datasets, particularly converging with FL under high data heterogeneity (Figure 3). We strongly believe that our empirical results can pave the way to a more nuanced theoretical understanding of this phenomenon as pointed out by the reviewer.
>
> **6) It is unclear to the reviewer whether Dproxy is contained in the local dataset.**
>
> → Yes, D_proxy is contained in the local training datasets for FL in all experiments. We will clarify it better in the final version.
>
> **7) Are error bars averaged across different random selection of Dproxys ?**
>
> → Yes, each seed corresponds to a sampling of a new D_proxy.

---

> > ### Comment · Reviewer_ne5T · 2023-11-20
> > **Response acknowledgment**
> >
> > The reviewer thanks the authors for their comments notably on acknowledging the similarities with late fusion. However the reviewer's major concern of the impact of Dproxy wasn't appropriately tackled nor did the authors commit to perform the proposed additional experiments. Therefore the reviewer will keep their initial scoring.

---

### Official Review · Reviewer_zo9y · 2023-10-29

**Soundness:** 2 fair
**Presentation:** 2 fair
**Contribution:** 2 fair
**Rating:** 3
**Confidence:** 3

**Summary:**

One-shot FL, i.e. aggregating models once at the end of federated learning, is known to be the best in terms of communication efficiency, but its performance is worse in general especially for heterogeneous data. The authors of this paper consider ensembles in the cross-silo heterogenous setting. They show experimentally that an aggregator based on a shallow neural nets significantly improve performance over one-shot FL, a method which they call FENS. It matches the performance of iterative FL and yet potentially uses 9x less communication. They report their results on various datasets and benchmarks such as CIFAR-10, SVHN, and FLamby.

**Strengths:**

- The usage of nonlinear aggregation is challenging and interesting, and it has some good potential. Finding a practical nonlinear aggregator method is an important direction to explore.
- The method particularly better on highly non-iid, which is a more challenging setting.
- The approach is intuitive and simple and easy to implement.
- This is an experimental paper, and I think that the experiment setup and results are explained very well.
- The experiments are reproducible, which is great. Unfortunately, the code is not shared in the supplementary materials.

**Weaknesses:**

- FENS requires proxy dataset, which might not always be available.
- An MLP aggregator implies a large dimension of the inputs, which in turn imply that the "shallow" neural net aggregator might have a large number of parameters. For example, if the client's model has 10K parameters (which is not a lot in cross-silo), and there are 10 clients (also not a lot), then the input dimension itself is 100K, so your aggregator for this simple case will have at the very least 1 billion parameters, which is quite a lot of parameters for some shallow neural net.
- It might be unfair to compare with methods with no trainable aggregator and no proxy dataset. Perhaps you can compare to a one-shot FL algorithm with a fine-tuned aggregator? You can consider an experiment where you train an MLP aggregator vs. tune the weights of the non-trainable aggregator (i.e. training a linear aggregator) on some proxy dataset. Then you can consider another experiment where you just average the models at the end and simply train the averaged model on the proxy dataset. You should show that using an MLP aggregator is indeed a better way to improve performance. Finally, you should consider the effect of the size of the proxy dataset on the performance because you might not always get a 10% cut of the original data publicly available.
- I’m not sure about experiment 4.2 and whether it is conclusive or not. You can tell that the increase is larger for FENS, but maybe that’s only because FedAvg was already closer to its top performance. The reason there seems to be a larger increase might be due to the efficiency of FedAvg at learning from fewer samples more than it is due to the efficiency of FENS at learning faster with more samples. Moreover, in this particular experiment, FENS seems to be performing strictly worse than FedAvg in this example, which is not a strong selling point of the method no matter how quickly it improves, especially since the improvement plateaus before it reaches FedAvg.

**Questions:**

- The authors need to explain the low performance on iid data in more details. It should still be good as this case is strictly easier. The trainable aggregator might be hurting performance in this case. Why is this the case? Perhaps it would be helpful to investigate this by running experiments similar to the ones proposed in the "Weaknesses" section above.

---

> ### Author Response · Authors · 2023-11-15
> **Response to Reviewer zo9y**
>
> We kindly thank the reviewer for their thorough evaluation and constructive feedback. We appreciate the time and effort you have dedicated to assessing our work and we are happy to provide more clarifications on some of the aspects below.
>
> **1) FENS requires proxy dataset, which might not always be available.**
>
> → This is indeed a limitation of our approach. However, more and more recent works are underscoring the importance and availability of D_proxy for various purposes [1].
>
> [1] J. Nguyen, et al., “Where to begin? on the impact of pre-training and initialization in federated learning,” in ICLR, 2023. Available: https://openreview.net/forum?id=Mpa3tRJFBb
>
> **2) An MLP aggregator implies a large dimension of the inputs. For example, if the client's model has 10K parameters (which is not a lot in cross-silo), and there are 10 clients (also not a lot), then the input dimension itself is 100K, so your aggregator for this simple case will have at the very least 1 billion parameters.**
>
> → We noticed that there might be a misunderstanding regarding the FENS aggregation function. In our task, we seek to aggregate the logits produced by trained models and not the model parameters. Hence, the input to the NN aggregator has a dimension of n_clients x n_classes (20 x 10 for CIFAR-10). Therefore our MLP with 1 hidden layer has less than 4500 trainable parameters. We will make it clearer in the final version.
>
> **3) It might be unfair to compare with methods with no trainable aggregator and no proxy dataset. Perhaps you can compare to a one-shot FL algorithm with a fine-tuned aggregator? You can consider an experiment where you train an MLP aggregator vs. tune the weights of the non-trainable aggregator (i.e. training a linear aggregator) on some proxy dataset. Then you can consider another experiment where you just average the models at the end and simply train the averaged model on the proxy dataset. Finally, you should consider the effect of the size of the proxy dataset.**
>
> → With reference to our clarification in the previous question, we note that the task of aggregating trained models instead of predictions is considerably more challenging [1, 2]. Hence, typical one-shot methods do not target model aggregation but instead use knowledge distillation for model fusion [3]. We compared against one such standard method called FedKD in Table 3 which uses a large public dataset at the server for knowledge distillation. Furthermore, simple averaging of trained models yields very poor performance due to client drift arising from heterogeneous data. This is reflected through the performance of one-round FedAvg in Table 3 where the accuracy remains poor even after explicit tuning of the number of local steps conducted before aggregation. However, as appropriately suggested by the reviewer, additional experiments with varying D_proxy size would be interesting and we plan on including them in the final version.
>
> [1] Yurochkin, Mikhail, et al. "Bayesian nonparametric federated learning of neural networks." ICML. PMLR, 2019.
>
> [2] Hongyi Wang, et al. “Federated learning with matched averaging”. In ICLR, 2020a. URL https://openreview.net/forum?id=BkluqlSFDS.
>
> [3] Gong, Xuan, et al. "Preserving privacy in federated learning with ensemble cross-domain knowledge distillation." Proceedings of the AAAI Conference on Artificial Intelligence. Vol. 36. No. 11. 2022.
>
> **4) I’m not sure about experiment 4.2 and whether it is conclusive or not. You can tell that the increase is larger for FENS, but maybe that’s only because FedAvg was already closer to its top performance. The reason there seems to be a larger increase might be due to the efficiency of FedAvg at learning from fewer samples more than it is due to the efficiency of FENS at learning faster with more samples.**
>
> → We fully agree with the explanation pointed out by the reviewer. Indeed, FedAvg is much more efficient in handling fewer data samples. However, this comes at a significant communication cost due to its iterative nature. Therefore, the goal of our experiments in section 4.2 was to identify scenarios when FENS can match FedAvg, which is a strictly stronger algorithm. We established that when data is non-IID and clients have sufficiently large local datasets, FENS can match FedAvg and avoid the iterative communication costs. However, this does not hold in the IID setting as we explain below.
>
> **5) The authors need to explain the low performance on iid data in more details. It should still be good as this case is strictly easier.**
>
> → As pointed out by the reviewer, the reason for the low performance actually stems from the trainable aggregator. Precisely, when the data is IID distributed, the logits produced by client models are very similar, thereby inducing additional difficulty for the aggregator in identifying patterns. FedAvg, in turn, bypasses this challenge through iteratively learning on the data and renders itself strictly better than FENS.

---

> > ### Comment · Reviewer_zo9y · 2023-11-20
> >
> > Thanks for your clarifications regarding the aggregation process. However, I still do not understand why FENS perform worse on iid data. There should be no pattern to discern across clients as the data are iid. In other words, each client would converge to the right model given sufficient data. In any case, there should be a way to mitigate FENS low performance on IID data. For example, at what level of heterogeneity would FENS start to show superior performance? Also, how would you do inference on this model? Would you have to aggregate logits every forward pass? It would have been great if you could have shared the code as well so that these questions can be easily answered from the implementation.

---

> > > ### Author Response · Authors · 2023-11-21
> > > **Response to Reviewer zo9y's comment**
> > >
> > > Thank you for your thoughtful engagement with our work, and we hope our clarifications below address your concerns.
> > >
> > > **However, I still do not understand why FENS perform worse on iid data. There should be no pattern to discern across clients as the data are iid. In other words, each client would converge to the right model given sufficient data. In any case, there should be a way to mitigate FENS low performance on IID data.**
> > >
> > > → Thank you for raising this point. We believe there are two contributing factors to FENS's lower performance on IID data:
> > >
> > > i) In general, no individual client's local baseline would match the performance of iterative FL, primarily because FENS clients learn solely from local data, while FL learns iteratively from the entire (distributed) dataset. It's conceivable that when each client has a substantial volume of local data, the performance might approach that of FL. However such a point seems to occur beyond the size of the datasets (FENS < FL even at full training fraction for IID in Fig. 3).
> > >
> > > ii) Additionally, when each client's local data is IID, the local models tend to produce similar logits. This similarity poses an additional learning challenge for the aggregator. In contrast, in non-IID cases, where local models generate significantly different logits, aggregation can more easily discern patterns. Thus, the ensemble's performance remains lower in the IID setting.
> > >
> > > In summary, we speculate that achieving parity with iterative FL in IID scenarios poses a challenge, as it would be surprising for one-shot FL to outperform or match iterative FL across all heterogeneity regimes.
> > >
> > > **For example, at what level of heterogeneity would FENS start to show superior performance?**
> > >
> > > → As we show in Figure 2, the performance of FENS starts matching FL as the data heterogeneity increases. Specifically, this happens at the alpha = 0.1.
> > >
> > > **Also, how would you do inference on this model? Would you have to aggregate logits every forward pass?**
> > >
> > > → Yes, FENS requires a forward pass through all the models in the ensemble followed by the aggregation of logits when inferring for an input sample. We explained the creation of the ensemble in Section 3.2 which might help clarify it better. If there are specific points that still appear unclear, we would be happy to provide additional details or discuss them further.
> > >
> > > **It would have been great if you could have shared the code as well so that these questions can be easily answered from the implementation.**
> > >
> > > → Thank you for your interest and we would have been delighted to share the code. However, in the spirit of providing a well-documented and user-friendly resource, we have decided to make the code available concurrently with the final version of the paper.

---

### Official Review · Reviewer_4Q6E · 2023-10-30

**Soundness:** 3 good
**Presentation:** 3 good
**Contribution:** 2 fair
**Rating:** 5
**Confidence:** 5

**Summary:**

This work proposed one-shot federated learning using an NN-based aggregator, which can significantly improve the performance of ensembles under high data heterogeneity. Extensive experiments were conducted to demonstrate the proposed method outperforms the baselines by a large margin.

**Strengths:**

1. Developed an NN-based aggregator, which can significantly improve the performance of ensembles under high data heterogeneity.

2. Conduct extensive experiments to show the proposed method achieve better accuracy than baselines.

**Weaknesses:**

1. Why does FEDCVAE-ENS (ResNet-8) have lower estimation accuracy than FEDCVAE-ENS (default CNN)? Do you use the decoder of ResNet-8 in the experiments?

2. It is unfair to compare the proposed approach to a data-free FEDCVAE-ENS.

3. The technical contribution is somewhat limited. This work only adopted a simple NN to aggregate the outputs.

**Questions:**

1. Why do you set the batch size to 16?

2. Why does FEDCVAE-ENS (ResNet-8) have lower estimation accuracy than FEDCVAE-ENS (default CNN)? Do you use the decoder of ResNet-8 in the experiments?

---

> ### Author Response · Authors · 2023-11-15
> **Response to Reviewer 4Q6E**
>
> We kindly thank the reviewer for their valuable feedback and questions. We appreciate the time and effort you have dedicated to assessing our work and we are happy to provide more clarifications on the questions raised.
>
> **1) Why does FEDCVAE-ENS (ResNet-8) have lower estimation accuracy than FEDCVAE-ENS (default CNN)? Do you use the decoder of ResNet-8 in the experiments?**
>
> → The FedCVAE-Ens method generates synthetic samples at the server using the decoders uploaded by the clients. The server then trains a classifier model from scratch on these samples. We used the same VAE architecture as the authors and experimented on two different classifier architectures – default CNN (as used by the authors) and ResNet-8 for fair comparison with other methods. We speculate two reasons for the lower estimation accuracy of ResNet-8 for the FedCVAE-Ens algorithm –
>
> (i) The ResNet-8 architecture introduces deeper and more complex residual connections compared to the default CNN. While this design can be advantageous for capturing intricate features in large datasets, it might lead to overfitting or inefficiency in the context of our specific task i.e. when learning on synthetic samples generated from the decoder.
>
> (ii) The CNN might be more suitable for the given choice of VAE architecture. We also contacted the authors to verify the correctness of our results on the CIFAR-10 task. They reported a sensitivity to the VAE architecture as one of the potential reasons for the overall low performance on CIFAR-10 in comparison to the stronger reported performance on MNIST.
>
> **2) It is unfair to compare the proposed approach to a data-free FEDCVAE-ENS.**
>
> → Although data-free, FedCVAE-Ens is one of the very few state-of-the-art methods dealing with data heterogeneity in the one-shot FL setting. Our comparison was for completeness i.e. to provide performance insights across both regimes – data-based and data-free.
>
> **3) The technical contribution is somewhat limited. This work only adopted a simple NN to aggregate the outputs.**
>
> → We appreciate the opportunity to address concerns regarding the perceived limitation of our technical contribution. Our primary focus is not merely on proposing a neural network aggregator but on a more fundamental exploration – assessing the viability of one-shot Federated Learning (FL) in comparison to iterative FL. While prior research in one-shot FL has shown promise, it has often struggled in the face of high data heterogeneity, lagging behind the performance of iterative approaches. In response to this challenge, we revisited ensemble techniques, conducting a thorough analysis across various aggregation functions in one-shot FL. Our comprehensive investigation revealed that the enhancement provided by the neural network aggregator is pivotal, enabling one-shot FL methods to achieve performance parity with iterative FL in highly heterogeneous settings—a scenario where these methods traditionally face limitations due to constrained communication. To the best of our knowledge, we are the first to demonstrate this equivalence. We believe that our empirical results can pave the way to more nuanced theoretical understanding of this phenomenon. We are grateful for the opportunity to clarify the significance of our work and its contribution to the broader field of Federated Learning. If there are specific areas that we can further elaborate on or address, we would be more than happy to do so.
>
> **4) Why do you set the batch size to 16?**
>
> → We adopt the standard practice of using a fixed batch size from relevant related work, for instance, 16 in [1]  and 20 in [2].
>
> [1] Xuan Gong, Abhishek Sharma, Srikrishna Karanam, Ziyan Wu, Terrence Chen, David Doermann, and Arun Innanje. Preserving privacy in federated learning with ensemble cross-domain knowledge distillation. Proceedings of the AAAI Conference on Artificial Intelligence, 36(11):11891–11899, Jun. 2022. doi: 10.1609/aaai.v36i11.21446. URL https://ojs.aaai.org/index.php/ AAAI/article/view/21446.
>
> [2] Sashank J. Reddi, Zachary Charles, Manzil Zaheer, Zachary Garrett, Keith Rush, Jakub Konecny, Sanjiv Kumar, and Hugh Brendan McMahan. Adaptive federated optimization. In International Conference on Learning Representations, 2021. URL https://openreview.net/forum? id=LkFG3lB13U5.

---

> > ### Comment · Reviewer_4Q6E · 2023-11-22
> > **Thanks for the response**
> >
> > Thanks for your response. I tend to keep my initial sore since the other reviewers also have concerns about the contribution of this work. However, I may change the score after the discussion among the reviewers.

---

### Official Review · Reviewer_hFH5 · 2023-11-10

**Soundness:** 3 good
**Presentation:** 3 good
**Contribution:** 2 fair
**Rating:** 5
**Confidence:** 3

**Summary:**

In this paper, authors focus on one-shot federated ensumbles in hetergenous cross-silo settings. One shot federated ensumbles is different from traditional federated learning as it only utlize one rounds of communication/aggregating of locally trained models. In general ensumble framework is used in such one-shot setting. Authors propose a new aggregator which is based on a shallow neural network and which is shown to siginificantly boost the perforamnce of ensembles under high data hetergeneity. In all experiments and ablation studies, such proposed method indeed shows large improvement over exisiting baselines.

**Strengths:**

1: One shot federated learning is indeed a very important research direction in FL as it totally avoids the communication cost during training and potentially avoids all network limitation (such as network safety and stability). I think any research in this direction will be valuable for the FL community.

2: The proposed method, i.e. using a shallow network as aggregator, is very simple and straight forward, which also induces little additional inference cost.

3: Extensive ablation studies and real life task settings are always welcomed.

4: Finally, a general strength for all ensemble methods is that they can support model hetergenorty and each client can utilize different size of model based on their computation capacity.

**Weaknesses:**

1: My major concern is that the novelty of the proposed method is very limited. Shallow network as aggregator has been widely used in other ensemble framework. More advanced methods such as using attention based aggregator have also been proposed. Thus, simply applying such method to FL setting is not a enough contribution (also I feel this also has been explored by previous papers).

2: It is also unclear what is the architecture of the proposed shallow network: does it only use the logits of all local models as input? Isn't it too limited? Why doesn't it utilize the information/embedding of input image? Again I think more research into the architecture of the aggregator is needed.

2: The model used in the experiments is very limited (ResNet-8). I feel it needs to include larger models and tested on larger dataset with more classes to really show its performance on hetergenous settings.

**Questions:**

Please see weakness section for more detailed questions.

Overall my major question is: what is the novelty of the proposed method compared with other ensemble method using network aggregator?

---

> ### Author Response · Authors · 2023-11-15
> **Response to Reviewer hFH5**
>
> We kindly thank the reviewer for their constructive and valuable feedback. We appreciate the time and effort you have dedicated to assessing our work and we are happy to address some of the concerns below.
>
> **1) My major concern is that the novelty of the proposed method is very limited. Shallow network as aggregator has been widely used in other ensemble framework. More advanced methods such as using attention based aggregator have also been proposed. Thus, simply applying such method to FL setting is not a enough contribution (also I feel this also has been explored by previous papers).**
>
> $\rightarrow$ We agree that the application of NN aggregator to aggregate predictions is not novel by itself. However, our contribution is not proposing the NN aggregator but understanding whether one-shot FL methods are capable of matching iterative FL. While previous work in one-shot FL has shown promising results, the methods have fallen considerably short of iterative FL and struggled under high data heterogeneity. To this end, we revisited ensemble techniques and carried out an in-depth analysis in one-shot FL, covering several aggregation functions. Our findings reveal that the boost provided by NN aggregator is sufficient for one-shot FL methods to match iterative FL in highly heterogeneous settings which is atypical for these methods due to limited communication.To the best of our knowledge, we are the first to show this performance equivalence between one-shot and iterative FL.
>
> **2) It is also unclear what is the architecture of the proposed shallow network: does it only use the logits of all local models as input? Isn't it too limited? Why doesn't it utilize the information/embedding of input image? Again I think more research into the architecture of the aggregator is needed.**
>
> → For the NN aggregator, we use a simple MLP comprising 1 hidden layer of 120 units and ReLu activation. The MLP is only fed with logits of the trained models as the input. Therefore, the input size is relatively small i.e. n_clients x n_classes (20 x 10 for CIFAR-10). We show that such a favorably sized NN can be efficiently and quickly trained by leveraging D_proxy data as the training data which is typically much smaller than the distributed D_train dataset i.e. |D_proxy| << |D_train|. We agree with the reviewer that more advanced aggregator architectures would be interesting to explore. However, our goal was not to present the best architecture but to understand the impact of a simple NN aggregator in heterogeneous setups. Our results demonstrate that even a simple NN aggregator can boost the performance significantly. Also, as pointed out by the reviewer, the use of image embeddings for better knowledge fusion has been previously studied in the traditional iterative FL [1] where the goal is considerably more challenging – model fusion – unlike prediction aggregation in our one-shot setting.
>
> [1] He, Chaoyang, Murali Annavaram, and Salman Avestimehr. "Group knowledge transfer: Federated learning of large CNNs at the edge." Advances in Neural Information Processing Systems 33 (2020): 14068-14080.
>
> **3) The model used in the experiments is very limited (ResNet-8). I feel it needs to include larger models and tested on larger dataset with more classes to really show its performance on heterogeneous settings.**
>
> → We adopted our choice of models from several previous works studying the same learning task [1,2,3]. Beyond ResNet-8, we also experimented with DeepMIL and EfficientNet (Table 4) in the FLamby benchmark experiments. While we believe that we have sufficiently assessed the performance of FENS, we will consider adding another dataset in the final version if the reviewer believes it can further enhance the assessment of FENS.
>
> [1] Lin, Tao, et al. "Ensemble distillation for robust model fusion in federated learning." Advances in Neural Information Processing Systems 33 (2020): 2351-2363.
>
> [2] He, Chaoyang, Murali Annavaram, and Salman Avestimehr. "Group knowledge transfer: Federated learning of large CNNs at the edge." Advances in Neural Information Processing Systems 33 (2020): 14068-14080.
>
> [3] Gong, Xuan, et al. "Preserving privacy in federated learning with ensemble cross-domain knowledge distillation." Proceedings of the AAAI Conference on Artificial Intelligence. Vol. 36. No. 11. 2022.

---

### Author Response · Authors · 2023-11-19

We would like to thank the reviewers once again for their valuable feedback. Since the discussion period is coming to an end, we would be happy to reply if the reviewers have any further questions.

---

### Meta-Review · Area_Chair_Rp3n · 2023-12-07

**Metareview:**

Summary:
This paper focuses on one-shot federated ensembles in heterogeneous cross-silo settings. One-shot federated ensembles differ from traditional federated learning as they only utilize one round of communication/aggregating of locally trained models. In general, an ensemble framework is used in such a one-shot setting. The authors propose a new aggregator based on a shallow neural network that is shown to boost the performance of ensembles under high data heterogeneity significantly. In all experiments and ablation studies, such a proposed method significantly improves over existing baselines.

Strengths:
+ The proposed method is straightforward.
+ Extensive ablation studies and real-life task settings are appreciated.

Weaknesses:
The novelty of the proposed method is very limited.
- More research into the architecture of the aggregator is needed.
- Experiments are limited and should be better understood.

**Justification For Why Not Higher Score:**

See above

**Justification For Why Not Lower Score:**

N/A

---

### Decision · Program_Chairs · 2024-01-16

Reject